# Optical charge injection and coherent control of a quantum-dot spin-qubit emitting at telecom wavelengths

Łukasz Dusanowski [1,4]✉, Cornelius Nawrath [2], Simone L. Portalupi [2], Michael Jetter[2], Tobias Huber[1], Sebastian Klembt [1], Peter Michler[2] & Sven Höfling [1,3]

Solid-state quantum emitters with manipulable spin-qubits are promising platforms for quantum communication applications. Although such light-matter interfaces could be realized in many systems only a few allow for light emission in the telecom bands necessary for long-distance quantum networks. Here, we propose and implement an optically active solid-state spin-qubit based on a hole confined in a single InAs/GaAs quantum dot grown on an InGaAs metamorphic buffer layer emitting photons in the C-band. We lift the hole spin-degeneracy using an external magnetic field and demonstrate hole injection, initialization, read-out and complete coherent control using picosecond optical pulses. These results showcase a solid-state spin-qubit platform compatible with preexisting optical fiber networks.

[1] Technische Physik and Würzburg-Dresden Cluster of Excellence ct.qmat, Physikalisches Institut and Wilhelm-Conrad-Röntgen-Research Center for Complex Material Systems, Am Hubland, University of Würzburg, Würzburg, Germany. [2] Institut für Halbleiteroptik und Funktionelle Grenzflächen (IHFG), Center for Integrated Quantum Science and Technology (IQST) and SCoPE, University of Stuttgart, Stuttgart, Germany. [3] SUPA, School of Physics and Astronomy, University of St Andrews, St Andrews, UK. [4] Present address: Department of Electrical and Computer Engineering, Princeton University, Princeton, NJ, USA. ✉email: lukaszd@princeton.edu

Long-distance quantum network technologies require reliable interfaces between stationary qubits and photons with frequencies in the telecom bands[1,2]. In this regard, spin degrees of freedom in solid-state quantum emitters are of particular interest due to the potential of realizing quantum entanglement between a confined spin and a propagating photon[3–6]. Spin-based photonics have been to date realized using a wide range of material systems. Each platform varies in the terms of electronic structure, emission wavelength, optical performance, spin-coherence, and integration capabilities with photonic devices[7,8]. Semiconductor quantum dots (QDs) for instance, show excellent optical properties[9,10], achieve spin-coherence times on the level of tens of microseconds[11,12], and allow for efficient generation of spin–photon[4–6,13] and spin–spin[14,15] entanglement. Another well-established spin-system, vacancy centers in diamond, while suffering intrinsically low emission efficiency into the zero-pho-non-line, show very long coherence times on the level of single seconds[16], and allowed for demonstration of spin–spin entanglement on the record distance of 1.3 km[17]. While diamond is still a rather challenging material for micro-processing, platforms based on defects in SiC show similar optical-performance with milliseconds spin-coherence times[18,19], and stand out by maturity of SiC and processing methods. Another promising class of spin-active emitters are rare-earth-ion doped crystals embedded in cavities[20–22]. Such systems might in principle achieve over seconds spin-coherence times[23], but require the usage of high-finesse cavities to enhance the intrinsically slow emission rates.

Among those many types of solid-state platforms, only a few systems demonstrated capability of directly interfacing spins with telecom-wavelength photons. Recently, spin-control has been demonstrated at 1220 nm wavelength in nitrogen-vacancy centers in SiC[24], at 1330 nm using ensemble of radiation T-damage centers in $^{28}$Si[25], and at 1550 nm in $Er^{3+}$ in $Y_2SiO_5$[21,22]. In particular, the latter system operates in the lowest-loss telecom C-band centered at 1550 nm, promising a clear advantage for long-distance quantum communication applications[2]. Emitters with transitions incompatible with fiber networks, such as defects in diamond, could potentially take advantage of quantum frequency conversion into telecom wavelengths[4,26]. While such an approach offers flexibility in terms of frequency matching of multiple emitters[27], it is intrinsically limited by the efficiency of the frequency conversion process and requires a considerable technological overhead.

Alternative systems that could potentially allow interfacing spins with telecom-wavelength photons are InAs QDs relying on InP substrate[28–30] or strain-relaxed layers in GaAs[31–33]. Spectroscopic studies of such systems identified already charged exciton complexes with intrinsic optical transitions in the C-band offering access to isolated electron and hole states[29,30,33,34].

In this work, we introduce a spin-qubit system based on a single hole confined in a InAs self-assembled QD based on the strain-relaxed GaAs emitting radiation directly in the telecom C-band. We demonstrate the full set of spin-control operations consisting of spin injection, initialization, read-out and coherent rotation. We perform all protocols at 76 MHz repetition rate, where the spin-rotation is achieved on the picosecond time-scale using detuned optical pulses. This demonstration provides a clear pathway toward direct spin–photon entanglement in the telecom C-band for applications in long-distance quantum communication. Thanks to the QDs' intrinsically short radiative lifetimes of around 1 ns, our system provides a direct alternative to $Er^{3+}$ ions or platforms relying on frequency conversion.

## Results

**Spin system under investigation.** The physical system used in our experiment is presented in Fig. 1a. It consists of InAs QDs

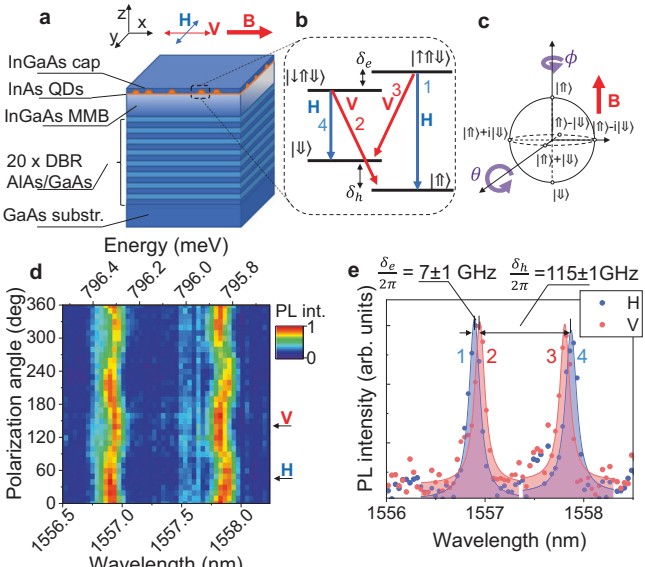

**Fig. 1 Quantum-dot spin platform. a** Sample layer structure and geometry used (not to scale). A single layer of InAs quantum dots is grown on an InGaAs metamorphic buffer (MMB) layer allowing for emission in the C-band telecom window. Bottom distributed Bragg reflectors (DBR, 20 pairs) and air top interface nominally form a 3-$\lambda$ cavity for increased photon collection efficiency. An external magnetic field (B) is applied along the x-axis (Voigt geometry). Emission collection and excitation is performed along the z-axis. Insets: orientation (x, y, z) and polarization convention (H, V) used. **b** Trion level structure in Voigt magnetic field with radiative transitions selection rules, as used in the experiment. **c** Bloch sphere representing the spin-qubit. **d** Color-coded map of the trion photoluminescence spectra as a function of detected polarization angle at 3 T magnetic field in Voigt configuration. **e** Trion photoluminescence spectra at B = 3 T at horizontal (H) and vertical (V) linear polarization.

grown on an InGaAs metamorphic buffer (MMB) layer, reducing the lattice mismatch between the QDs and the underneath material[35]. This results in an emission wavelength in the telecom C-band. In addition, the QDs and the MMB are placed on a near lattice-matched AlAs/GaAs distributed Bragg reflector (DBR) that significantly enhances the photon extraction efficiency (see Methods). For optical experiments, we cool-down the sample to 1.6 K temperature and focus on a single QD by using a confocal microscope. Under nonresonant optical excitation, sharp emission lines are identified in photoluminescence spectra stemming from radiative recombination of various excitonic complexes. In particular, positively charged trions formed by a single electron and two holes are identified as dominant transitions in agreement with previous theoretical and experimental studies[36]. Consequently, in this work, we focus on an isolated emitter with an optical transition between a positively charged exciton and a ground hole state (see Supplementary Note 2). The benefit of using the hole state over the electron is hole's smaller hyperfine interaction strength with nuclear spins due to the hole's p-like symmetry[37–43] and thus potentially better hole coherence.

Under a magnetic field oriented in Voigt geometry (perpendicular to the growth direction/optical axis), two $\Lambda$-systems are formed from an initial two-level system, where each optically active trion state $|\downarrow\Uparrow\Downarrow\rangle/|\uparrow\Uparrow\Downarrow\rangle$ is coupled to each of two ground hole spin states $|\Downarrow\rangle$ and $|\Uparrow\rangle$. The optical selection rules are indicated in Fig. 1b. In this work, we define a qubit as a quantum-dot hole spin-state $cos(\theta/2)|\Downarrow\rangle + e^{i\phi}sin(\theta/2)|\Uparrow\rangle$, which can be schematically represented on the Bloch sphere, where $\theta$ and $\phi$ are spherical co-ordinates as shown in Fig. 1c. The north and south

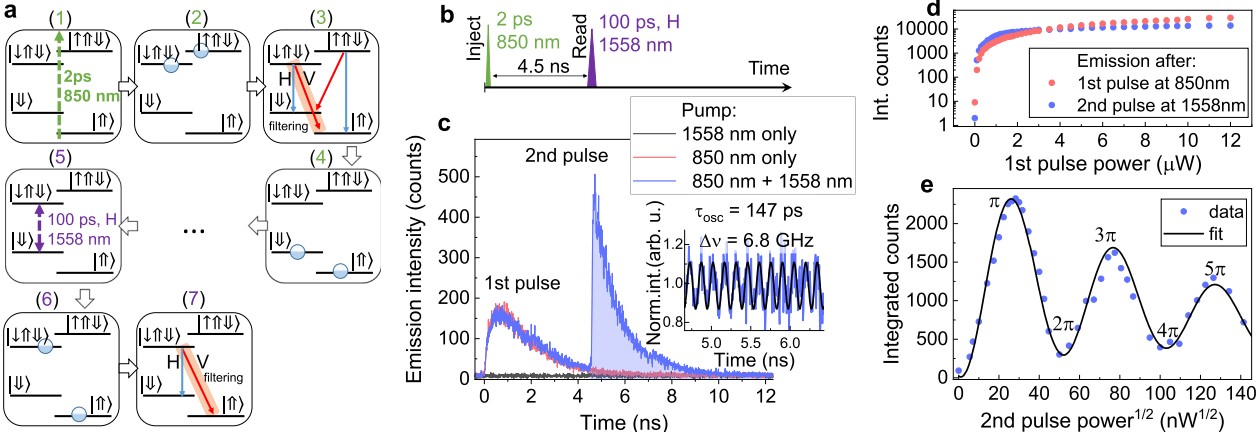

**Fig. 2 Charge injection and coherent control of trion state. a** Schematic description of optical charge injection (1–4) and coherent control of trion state (5–7) used in the experiment. First, (1) a nonresonant pulse excites carriers in the semiconductor which then relax (2) forming a trion state with random spin-configuration. Next, (3) the trion recombines radiatively (4) leaving the QD with a charge in a random $|\Downarrow\rangle$ or $|\Uparrow\rangle$ state. (5) The second pulse at H polarization resonant with $|\Downarrow\rangle$-$|\downarrow\Uparrow\Downarrow\rangle$ transition excites the system into (6) $|\downarrow\Uparrow\Downarrow\rangle$ trion state which can then recombine radiatively emitting an H or V-polarized photon. In the experiment the V-polarized $|\downarrow\Uparrow\Downarrow\rangle$-$|\Uparrow\rangle$ transition is filtered spectrally and detected. **b** The excitation pulse sequence used in the experiment. The first pulse at 850 nm injects the carrier into the QD and a second pulse at 1558 nm resonantly excites the system into the $|\downarrow\Uparrow\Downarrow\rangle$ state. **c** Time-trace of $|\downarrow\Uparrow\Downarrow\rangle$-$|\Uparrow\rangle$ emission after nonresonant pulse only (red curve), resonant pulse only (black curve) and both pulses (blue curve). Emission after resonant pulse exhibits slight amplitude oscillations with a period of 147 ± 1 ps corresponding to the $|\downarrow\Uparrow\Downarrow\rangle$-$|\uparrow\Uparrow\Downarrow\rangle$ states splitting. Inset: Intensity oscillations recovered from the time-trace. **d** Integrated emission counts after 1st (nonresonant) and 2nd (resonant) excitation pulse in the function of 1st (nonresonant) pulse power. **e** Integrated emission counts after 2nd (resonant) pulse in the function of the square root of the 2nd (resonant) pulse power. The emission exhibits clear Rabi oscillations up to $5\pi$ pulse area.

pole represent the pure $|\Downarrow\rangle$ and $|\Uparrow\rangle$ states. Access to any arbitrary Bloch vector on the sphere requires the ability of spin rotation along two perpendicular axes to control both $\theta$ and $\phi$ angle.

In Fig. 1d polarization dependence of the preselected trion photoluminescence at 3 T magnetic field in Voigt configuration is shown. The emission spectrum shows a four-fold splitting, with polarized transitions consistent with selection rules shown in Fig. 1b. By fitting the emission peaks at vertical (V) and horizontal (H) polarization (as defined in Fig. 1) we recover the trion and hole splitting of 7 ± 1 and 115 ± 1 GHz, respectively. Systematic investigation of the peaks in the function of the magnetic field reveal the ground spin-state and trion spin-state g-factors of 2.7 and 0.16, respectively (see Supplementary Note 3).

**Optical charge injection**. Within the current study, we utilize a nominally undoped QD sample, thus most of the QDs are uncharged. Interestingly, trion transitions are visible under nonresonant pumping, suggesting that photo-excited electrons and holes relax with different rates predominantly forming positive trion states. This, in turn, could be utilized to load the QD with a single charge, since directly after trion recombination the QD is left in the ground hole state. Moreover, since trions can recombine with equal probability into both spin states, such a charge injection scheme randomizes the initial spin-configuration serving additionally as an active-spin-reset.

To experimentally demonstrate the optical charge injection, we perform a two-color pulse experiment as schematically described in Fig. 2a. First, we excite a QD with a 2 ps width nonresonant pulse at 850 nm to form a trion (1–2). Upon radiative recombination (3), the QD is left with single hole in random $|\Downarrow\rangle$ or $|\Uparrow\rangle$ spin-configuration (4). After 4.5 ns we send another H-polarized pulse with 100 ps width and central wavelength of 1558 nm, which is resonant with the $|\Downarrow\rangle$-$|\downarrow\Uparrow\Downarrow\rangle$ transition (5). Taken, that a charge was indeed loaded into the QD, the second pulse should excite the system into the $|\downarrow\Uparrow\Downarrow\rangle$ trion state (6),

which will then recombine radiatively (7). By spectrally filtering the diagonal V-polarized transition $|\downarrow\Uparrow\Downarrow\rangle$-$|\Uparrow\rangle$ and performing a time-resolved measurement, we can trace the QD emission dynamics following both pulses, as shown in Fig. 2c. Emission decay after the first pulse exhibits a mono-exponential decay with a time constant of 1.4 ns, as typically observed for such type of QDs[35]. Emission after the second pulse exhibits a similar mono-exponential decay, however, with a slight amplitude oscillation with a period of 147 ± 1 ps as indicated in the Fig. 2c inset. This oscillations period corresponds to a 6.80 ± 0.05 GHz frequency which is in good agreement with trion state splitting of 7 ± 1 GHz and is a fingerprint of coherent excitation. Interestingly, when the sample is excited with the resonant pulse only, emission from the QD is not observed (black curve). This allows recording the reference background time-trace related to the scattered laser. By integration of the counts corresponding to the first or second decay on the graph (blued area), we can plot the time-integrated intensity of the QD emission after each pulse. In Fig. 2d the emission intensity after each excitation pulse is plotted as a function of the first (charge injection) pulse power. Clear intensity increase and saturation at around 1 μW are observed for both decays. In Fig. 2e we plot the emission after the second (resonant) pulse as a function of the square root of the second pulse power. In this case, clear oscillatory behavior is observed related to coherent Rabi rotation of trion state population. We can observe rotation up to $5\pi$-pulse area, which is clear evidence of coherent trion control.

**Spin initialization**. To initialize the spin into the desired $|\Downarrow\rangle$/$|\Uparrow\rangle$ state we use optical pumping[44,45]. For that a continuous wave (cw) laser at H polarization is tuned into resonance with the transition $|\Uparrow\rangle$-$|\uparrow\Uparrow\Downarrow\rangle$/$|\Downarrow\rangle$-$|\downarrow\Uparrow\Downarrow\rangle$ (brown arrows in Fig. 3a, b inset). The fluorescence is monitored at the corresponding V-polarized transitions $|\uparrow\Uparrow\Downarrow\rangle$-$|\Downarrow\rangle$/$|\downarrow\Uparrow\Downarrow\rangle$-$|\Uparrow\rangle$. A significant increase of the emission intensity is observed when the resonance with the given transition is established as shown in Fig. 3a, b.

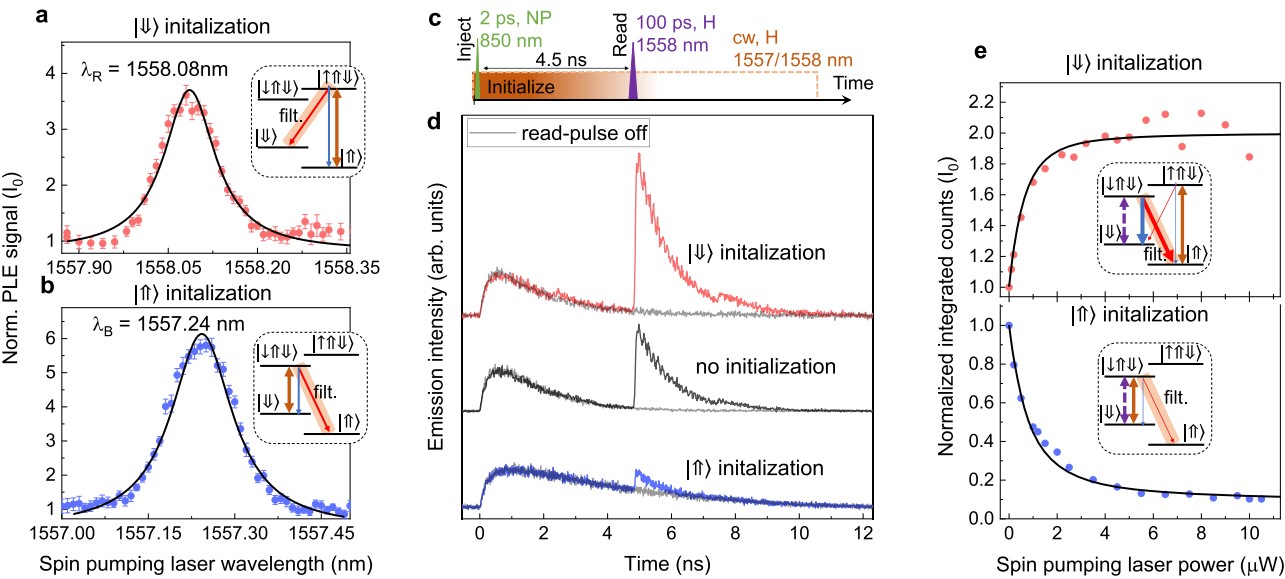

**Fig. 3 Spin initialization by optical pumping. a, b** The emission intensity of $|\uparrow\Uparrow\Downarrow\rangle$-$|\Downarrow\rangle$ and $|\downarrow\Uparrow\Downarrow\rangle$-$|\Uparrow\rangle$ transition as a function of spin-pumping laser wavelength scanned along the $|\uparrow\Uparrow\Downarrow\rangle$-$|\Uparrow\rangle$ and $|\downarrow\Uparrow\Downarrow\rangle$-$|\Downarrow\rangle$ transition, respectively. The signal is normalized to intensity in the absence of spin-pumping laser. Insets: Trion level schemes with relevant transitions. **c** Pulse sequence used for demonstration of $|\Uparrow\rangle$ and $|\Downarrow\rangle$ spin-initialization and subsequent spin read-out. First (nonresonant) pulse injects the carriers into the QD with random spin configuration. Then the cw spin-pumping laser initializes the system into $|\Uparrow\rangle$ and $|\Downarrow\rangle$. After 4.5 ns an H-polarized pulse resonant with $|\Downarrow\rangle$-$|\downarrow\Uparrow\Downarrow\rangle$ probes the $|\Downarrow\rangle$ state by exciting it into $|\downarrow\Uparrow\Downarrow\rangle$. **d** Time-trace of $|\downarrow\Uparrow\Downarrow\rangle$-$|\Uparrow\rangle$ emission demonstrating successful $|\Downarrow\rangle$ and $|\Uparrow\rangle$ initialization visible by increased and decreased emission intensity after a second (probe) pulse. **e** The emission intensity of $|\downarrow\Uparrow\Downarrow\rangle$-$|\Uparrow\rangle$ transition as a function of spin-pumping laser power normalized to the intensity observed in the absence of spin pumping. The data are fitted with a 4-level rate equation model. Insets: Trion level schemes with relevant transitions.

After a charge is injected into the QD by the nonresonant pulse, the cw pumping laser re-excites the $|\Uparrow\rangle$ state into $|\uparrow\Uparrow\Downarrow\rangle$, until the event of V-polarized emission, which will initialize the system into $|\Downarrow\rangle$. Analogously, by optical pumping of the transition $|\Downarrow\rangle$-$|\downarrow\Uparrow\Downarrow\rangle$, the system can be initialized in the $|\Uparrow\rangle$ state. Since the probability of V and H-polarized fluorescence is equal, the initialization process will require few emission cycles to reach high yield. Based on our simulations, we expect that within 4.5 ns (6 ns) initialization time we can reach 96% (99%) spin-preparation precision (see Supplementary Note 4).

To read the spin state in the QD we send a subsequent H-polarized pulse resonant with $|\Downarrow\rangle$-$|\downarrow\Uparrow\Downarrow\rangle$, which probes the $|\Downarrow\rangle$ state population (pulse sequence shown in Fig. 3c). In the case of the $|\Downarrow\rangle$ initialization, around two-fold enhancement of the emission intensity can be observed, while in the case of $|\Uparrow\rangle$ initialization clear suppression is visible as shown in Fig. 3d. In the case of $|\Uparrow\rangle$ pumping a re-excitation process is visible directly in the time-trace as a prolongation of the emission decay after the nonresonant pulse. In Fig. 3e integrated counts corresponding to the read-out pulse are plotted as a function of the spin-pumping laser power in case of the $|\Downarrow\rangle$ and $|\Uparrow\rangle$ initialization, respectively. For the $|\Downarrow\rangle$ pumping, the emission intensity reaches double the counts observable in case of no initialization, which is related to the increase of the $|\Downarrow\rangle$ state preparation probability from 50% to almost 100%. In analogy, in the case of $|\Uparrow\rangle$ pumping, significant suppression of the signal is observed. For spin-pumping powers above 10 μW, a spin-preparation precision of 94.5 ± 0.5% is achieved for 4.5 ns optical pumping time. This could be further improved by increasing the delay between charge injection and spin-read-out pulses.

**Coherent control of the hole spin.** For coherent rotation of the hole spin-state to any desired point on the Bloch sphere we

employ detuned circularly polarized laser pulses[46]. If the laser detuning $\Delta$ is much larger than the splitting between the hole ground states $\delta_h$, the net effect on the double $\Lambda$ system is such that the population of the spin state should oscillate with an effective Rabi frequency $\Omega_{eff}$, without physically exciting the system into the trion state. By keeping the pulse length constant and by controlling its power, it is possible to coherently rotate the spin state between $|\Downarrow\rangle$ and $|\Uparrow\rangle$. The rotation angle $\theta$ will be in such case proportional to the square root of the detuned laser power and inversely proportional to the detuning[46].

For spin-control experiments we apply a magnetic field of 1.5 T, decreasing the hole spin splitting to 57 GHz and increasing the Larmor precession period to around 17.5 ps. First, we send a nonresonant 850 nm charge injection pulse and initialize the system into the $|\Uparrow\rangle$ state by optical pumping. Then, after around 6 ns we send a circularly polarized ~400 GHz red detuned spin-rotation pulse with a length of 4 ps. To read if the spin was rotated from $|\Uparrow\rangle$ to $|\Downarrow\rangle$ state, a cw spin-pumping laser excites the system into the trion state. Following radiative recombination of a V or H-polarized photon signalizes the spin-rotation. The respective pulse sequence used is shown in Fig. 4b. By monitoring the emission evolution of the V-polarized $|\downarrow\Uparrow\Downarrow\rangle$-$|\Uparrow\rangle$ transition we can detect the spin-rotation events as shown in the time-trace in Fig. 4c. We note here that pumping powers required for spin rotation are quite significant such that the scattered laser suppression is challenging. To take into account the non-filtered laser contribution we subtract the background by recording reference time-trace with the initial charge injection-pulse switched off. In Fig. 4d we plot the emission signal after the spin-rotation pulse as a function of the square root of this pulse power (blue data points). Clear oscillatory behavior can be observed, related to Rabi rotations up to $3\pi$ angle. The evolution of the Bloch vector trajectory by angle $\theta$ during this process is schematically shown in the Fig. 4d inset. To estimate the off-resonant incoherent trion excitation contribution due to the

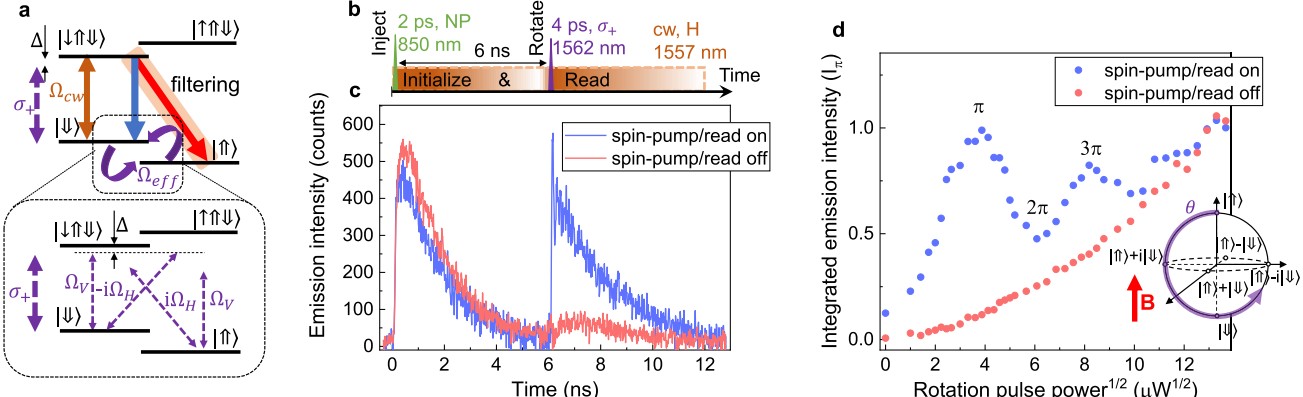

**Fig. 4 Optical coherent control of spin. a** Trion level structure in Voigt magnetic field geometry with transitions relevant for the spin-initialization, rotation, and read-out. Inset: For the spin rotation a stimulated Raman transition using circularly polarized pulses is used which in case of small detuning can be effectively reduced to spin-rotation of a two-level system. **b** Pulse sequence used to coherently rotate the spin. The first pulse injects the charge into the QD and a cw spin-pumping laser initializes the spin into the $|\Uparrow\rangle$ state. After 6 ns a circularly polarized Raman spin-rotation pulse is applied detuned by ~400 GHz from the trion transition. Immediately after spin-rotation into the $|\Downarrow\rangle$ state, the cw laser excites the system into $|\downarrow\Uparrow\Downarrow\rangle$ and a photon is emitted. **c** Emission time trace of the $|\downarrow\Uparrow\Downarrow\rangle$-$|\Uparrow\rangle$ transition in the presence and absence of the cw spin-pumping/read-out laser. **d** Emission intensity after the spin-rotation pulse as a function of the square root of the rotation pulse power. Clear Rabi oscillations are visible up to $3\pi$ demonstrating coherent control. Inset: Bloch sphere representation of the hole qubit evolution with the rotation pulse power.

rotation pulse, we record the emission time-trace with the cw spin-pumping laser switched off (red curve Fig. 4c). For the rotation powers above 100 μW, the incoherent part of the emission becomes dominant over the coherent part. The oscillation amplitude damping is most likely related to the combination of a few effects, such as finite rotation pulse length in respect to Larmor period (4 vs. 17.5 ps), presence of cw spin-pumping/read-out laser during the spin rotation and slight tilt of the rotation vector due to the QD asymmetry and non-perfect circular polarization of the rotation pulse.

**Ramsey interference and full coherent control**. To access all the qubit states on the Bloch sphere, spin rotation by a second axis is required. For that, we utilize the inherent Larmor precession of the spin state around the applied magnetic field[46]. This can be probed by the means of Ramsey interferometry. For that $|\Uparrow\rangle$ population is probed after two $\pi/2$ rotation pulses separated by the variable time delay $\tau$. The pulse sequence used and the corresponding evolution of the spin-state trajectory on the Bloch sphere are shown in Fig. 5a. After a first $\pi/2$ rotation pulse, the hole spin is rotated from the north pole to the equator, where it is allowed to freely precess around the external magnetic field with a Larmor period defined by $|\Downarrow\rangle$-$|\Uparrow\rangle$ splitting. By delaying a second $\pi/2$ pulse by an amount $\tau$, Ramsey fringes are observed. In Fig. 5b Ramsey interference for a pair $\pi/2$ pulses is shown, demonstrating integrated emission counts as a function of the time delay $\tau$ between the pulses. The data are fitted with a damped sine function exhibiting a Larmor period of $17.44 \pm 0.03$ ps ($57.3 \pm 0.2$ GHz hole splitting). Additionally, in Fig. 5c the amplitude of the Ramsey fringes, as a function of the delay time between the pulses is shown. The data are fitted by an exponential decay, revealing the inhomogeneous dephasing time $T_2^*$ of $240 \pm 30$ ps. While typically $T_2^*$ on the level of nanoseconds is observed in QDs[11,12,40], we relate the $T_2^*$ value recorded in our case, to the presence of the cw pumping between the rotation pulses[46], thus, $T_2^*$ could be potentially increased by temporal modulation of the pumping laser. The fringe contrast of the first Ramsey period reaches the level of 85% (after correcting for incoherent contribution).

Finally, to access any arbitrary state on the Bloch sphere, we demonstrate simultaneous control of $\theta$ and $\phi$ rotation angles by

adjusting the power of the pulses and delay time $\tau$ between them, using a pulse sequence as shown in Fig. 5d. In Fig. 5e we plot the QD integrated emission counts as a function of rotation pulse area—$\theta$ and delay time between the pulses $\tau$. For $\theta$ increasing from 0 to $\pi/2$ the Ramsey fringe amplitude rises and achieves a maximum value at $\pi/2$. When $\theta$ is further increased from $\pi/2$ to $\pi$, the fringe amplitude decreases and a phase shift is observed. Ramsey interference fringes recorded for $\pi/2$ and $\pi$ pulse areas are in clear anti-phase as shown in Fig. 5f. At $\theta = \pi$ the oscillations are supposed to vanish completely, however, some small signal is remaining. We relate this to the non-negligible spin Larmor precession with respect to the effective Rabi rotation frequency (nonzero pulse length, the QD asymmetry). Access to $\theta$ rotation in the range of 0 to $\pi$ and $\phi$ of 0 to $2\pi$ (one Larmor period) allows to explore the entire surface of the Bloch sphere and constitutes full coherent control of the spin-qubit. Taking into account the rotation pulse length and Larmor precession period, a single gate operation within our system could be performed within a 26 ps time window.

## Discussion

In conclusion, we have demonstrated optical spin injection, initialization, read-out and full coherent control of a spin-qubit system based on a hole confined in a single InAs/GaAs QD grown on a MMB layer. The measurements performed highlight this system as particularly promising for long-distance quantum network applications thanks to the possibility of interfacing spins with photons directly at C-band telecom frequencies. The QD-spin coherent control was achieved using spin rotations by two perpendicular axes using a detuned pair of picosecond pulses. Furthermore, Ramsey interferometry allowed us to probe the spin dephasing time, which in our case was limited to 240 ps due to the presence of optical pumping during the spin-rotation sequence. Regular InAs/GaAs QD systems operating at shorter wavelengths and driven by modulated initialization/read-out sequences have shown the $T_2^*$ times of over few ns[11,12,40].

While the intrinsic coherence limitations of the spin system investigated in this work remain to be revealed in further studies, the presence of strain-relaxed layers suggest that it could outperform the typical well-established InAs/GaAs QDs. It was shown that reducing the lattice strain in semiconductor QDs

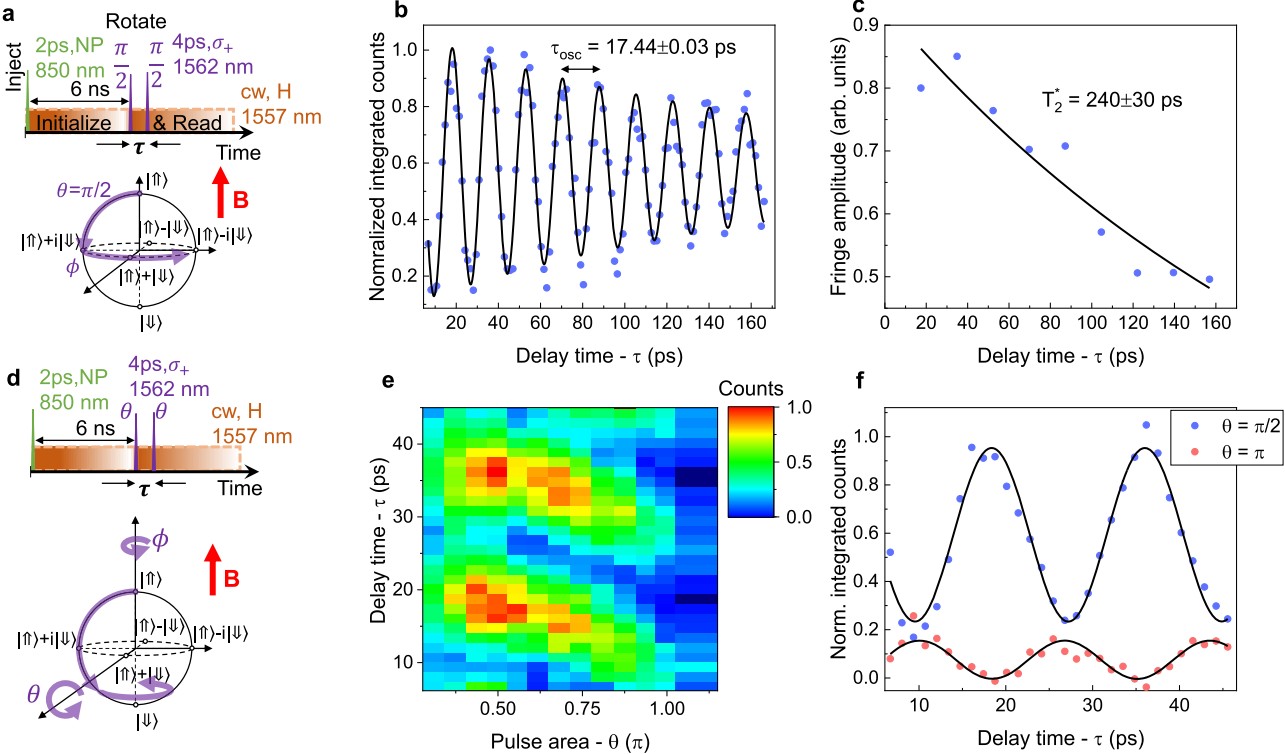

**Fig. 5 Ramsey interference and full coherent control. a** Pulse sequence used for Ramsey interference experiment. Around 6 ns after the charge injection pulse, two Raman spin-rotation pulses with $\pi/2$ area and time delay $\tau$ are applied. The evolution of the spin-qubit state in such case is schematically represented on the Bloch sphere below. After a first $\pi/2$ rotation pulse, the hole spin is rotated from the north pole to the equator, where it is allowed to freely precess around the external magnetic field B with a Larmor period defined by the $|\Downarrow\rangle$-$|\Uparrow\rangle$ splitting. By delaying a second $\pi/2$ pulse by $\tau$, Ramsey fringes are observed. **b** Ramsey interference for a pair of $\pi/2$ pulses, showing integrated emission counts as a function of the time delay between the pulses. The data are fitted with a damped sine function exhibiting a Larmor period of 17.44 ± 0.03 ps. **c** The amplitude of the Ramsey fringes, as a function of the delay time between the pulses $\tau$. Data are fitted by an exponential decay with a time constant of 240 ± 30 ps. **d** Pulse sequence used for simultaneous control of $\theta$ and $\phi$ spin rotation. By changing both the delay $\tau$ and the pulse power, the entire surface of the Bloch sphere can be explored. **e** Demonstration of full coherent control of the spin-qubit. Integrated emission counts as a function of the rotation pulse area $\theta$ and delay time between the pulses $\tau$. **f** Ramsey interference for a pair of $\pi/2$ and $\pi$-area pulses.

increases the spin-coherence by limiting the electric quadrupolar effects which play a dominant role in hole hyperfine interaction with nuclear spins[47,48]. In case of the hole qubits, less strain translates also into smaller light-hole admixture, which was shown to limit the spin-hole lifetime[49,50] and coherence time[48].

Generating the spin–photon and spin–spin entanglement with the current device architecture should be realistically achieved in the near future. Thanks to mature GaAs fabrication technology, our QD-spin system could be further combined with cavities and other photonic structures. This in turn might allow increasing the photon collection efficiency, speed up the spin-initialization process and achieve high optical transition cyclicity[51] for single-shot read-out. Further performance improvement could be achieved by utilizing an n–i–p diode structure, which was shown to allow trapping the charge state via Coulomb blockade, tuning the emission frequency using gate voltage and reducing the charge noise[50]. These advances combined with emission in the telecom wavelengths will be of immediate relevance for long-distance quantum networks technologies based on optical fibers or satellites.

## Methods

**Sample description.** The QD sample used in this study was grown by metalorganic vapor phase epitaxy (MOVPE) on (100) GaAs substrate. First, the 200 nm buffer layer of GaAs was deposited, followed by 20 alternating DBR pairs of AlAs/GaAs with thickness 134.4 and 114.6 nm, respectively. Next, a 1080 nm thick InGaAs metamorphic buffer (MMB) layer was grown by linearly increasing the In flux over the time from 0 to 17 µmol/min. The indium concentration increase with

the thickness in the MMB allows increasing the relaxation of the lattice constant. Finally, InAs material was deposited to form the QD layer, following capping with 220 nm of $In_{0.25}Ga_{0.75}As$. The bottom DBR and the top InGaAs–air interface nominally form a 3-$\lambda$ cavity optimized at 1550 nm and with 100 nm stop-band width. The QD layer is placed in the cavity mode anti-node for increased photon extraction efficiency from the top. The schematic sample layer structure is shown in Fig. 1a. The QD areal density varies on the sample position and is estimated to be $10^7$ cm$^{-2}$. More details regarding the sample fabrication process can be found in ref. [35].

**Experimental setup.** The sample is cooled down to a temperature of 1.6 K inside a closed-cycle liquid helium cryostat (Attodry 2100). The sample is placed in a dipstick cage-system, in which helium gas acts as an exchange gas. Superconducting magnet surrounding the sample chamber is used to apply a magnetic field up to 8 T. The optical part of the setup consists of a home-built confocal microscope mounted on top of the cryostat, and an apochromatic lens (NA = 0.68) inside the sample chamber. The sample is mounted on piezoelectric stages (Attocube) allowing to position the sample with respect to the lens. Nonresonant excitation of the QDs is performed using a mode-locked Ti:Sa pulsed laser (Coherent Mira) at 850 nm and 76 MHz repetition rate (2 ps pulse width). For resonant excitation, two lasers are used: (i) continuous wave tunable diode laser at 1550–1570 nm (iBlue ModBox) and (ii) tunable Optical Parametric Oscillator (Coherent OPO-X) pumped by the Ti:Sa laser at 850 nm. For the Ramsey interference experiments, a laser beam was passed through the free-space Mach-Zender interferometer where the time delay between the pulses is controlled via a retro-reflector mounted on the motorized translation stage. The fluorescence signal is collected through the same lens. A 1200 nm long-pass filter in the confocal microscope filters out the 850 nm laser excitation. Furthermore, polarization optics is setup in cross-polarization configuration for resonant laser suppression. Additional suppression of the laser is achieved by spatial filtering using single-mode fiber. Emission is then sent to a 75 cm focal length monochromator where it could be resolved spectrally and imaged on the array detector (Princeton Instruments

NIRvana) or filtered spectrally and coupled to superconducting single-photon counting detector (Single Quantum EOS) with a time resolution better than 20 ps. The signal from the single-photon counting detectors is sent to a time-tagger module (Pico Harp 300) triggered by the laser. The total timing resolution of the setup (instrumental response function fwhm) is around 30 ps. The experimental setup scheme can be found in the Supplementary Note 1.

**Data analysis**. The non-filtered laser contribution to the time-resolved histograms (visible under high pumping powers) is subtracted from the data by recording the reference histograms under identical conditions but with the charge injection laser (850 nm) switched off. Data shown in Fig. 5b, c, e, f are corrected for the incoherent emission contribution, decreasing the Ramsey fringes contrast from 85% to 72% under $\pi/2$ driving. In Fig. 3d there are two very small intensity peaks visible at 2.5 and 7.7 ns time. Those artefacts are related nonideal 76 MHz read-out laser pulse sequence, consisting of the main pulse and two extra satellite pulses at ±3 ns with around two orders of magnitude lower intensity.

## Data availability
The data that support the findings of this study are available from the corresponding author on reasonable request.

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

## Acknowledgements
We thank Yu-Ming He and Chao-Yang Lu for fruitful discussions at the beginning stages of this study. This work was funded by the Deutsche Forschungsgemeinschaft (DFG, German Research Foundation)-INST 93/932-1 FUGG. We acknowledge financial

support by the German Ministry of Education and Research (BMBF) within the project "Q.Link.X" (FKZ: 16KIS0871 and 16KIS0862). We are furthermore grateful for the support by the State of Bavaria.

## Author contributions
C.N., M.J., S.L.P., and P.M. designed and provided the wafer. Ł.D. established an experimental setup, carried out the optical experiments, analyzed, and interpreted the experimental data. T.H. and S.K. helped with the project administration and acquiring the funding. S.H. and P.M. guided the work. Ł.D. wrote the paper with input from all authors.

## Funding

## Competing interests
The authors declare no competing interests.
