## [Peer Review File · Nature Communications]

REVIEWER COMMENTS

Reviewer #1 (Remarks to the Author):

Solid-state spin qubits, which can be manipulated with C-band photons, have been expected as one of key elements for long-distance quantum networks. This report, demonstrating the all-optical coherent control of a spin-qubit in a quantum dot in the C-band, will provide a route towards the goal. The manuscript is well organized and describes the experiments the authors performed and the relevant discussions clearly.

However, based on the following facts, I think that the impact of the current work is not sufficiently strong as a paper in Nature Communications.

1. As the authors cite, coherent manipulation of spins in semiconductor QDs has been reported previously in many papers. Although the operation wavelengths are at around 1 μ m, out of the C band, the fundamental schemes for the spin manipulation applied in the present work have been already discussed in the previous reports.

2. The technology enabling the demonstration, the crystal growth technology for C-band QDs on GaAs substrate, has been reported in ref [28].

On the other hand, I believe that it is worth reporting the sophisticated and high-quality experiments somewhere else.

Other specific comments are listed below.

1. In this study, hole spins are utilized for the demonstrations. Why did the authors choose X⁺ not X⁻?

2. What are advantages to use hole spins instead of electron spins?

Reviewer #2 (Remarks to the Author):

This paper presents a set of measurements on a new type of quantum-dot (QD) spin qubit, which naturally emits light in the telecom range. Historically, InAs QDs typically emit at shorter wavelengths, but emission into the telecom range is a major benefit for quantum networks. Some of

the authors of the present work developed a new type of InAs QD that emits around 1550nm a few years ago, and the present paper shows that this system can be used as a spin qubit.

This paper demonstrates coherence of the charge and spin dynamics using a set of “conventional” measurements that are typical for regular InAs electron-spin QDs. On the whole, the data are convincing and, I think this work has the potential to be a major step forward. I have two main comments for the authors:

1. At various points the authors mention the fidelities of different operations of their spin qubit, even though they have not performed state or process tomography. I think I understand their intended meaning, and I know that language of this sort may be common in the InAs QD literature, especially the older works. However, I think it best to reserve the term “fidelity” for cases when state or process tomography have been performed, especially in today’s world. I think the authors are not trying to claim record fidelities, so this should not be so challenging to fix. They could for example, just omit their claims of fidelities and leave the visibilities specified.

2. The data presented show that the system works reasonably well as a qubit, but I was hoping they could convince me that it would be a really good qubit. Some questions that come to mind include:

a. Do the authors think it is possible to achieve higher degrees of spin-photon entanglement (at telecom wavelengths) with their telecom QDs than with regular InAs QDs and wavelength converters?

b. What challenges do they expect to encounter, and how could these be overcome?

c. Does the hole nature of this qubit confer any advantages (or disadvantages) with respect to electron-spin QDs?

d. What about the yield of their QDs with respect to regular InAs QDs?

e. It seems like the frequency shifts between the H and V lines (Fig. 1 d) are smaller than in conventional InAs QDs—does this make a difference to the qubit performance, especially in cases where readout fidelity is important, like teleportation?

Reviewer #3 (Remarks to the Author):

Dusanowski et al describe the first demonstration of spin control and manipulation with an optical interface in the telecom C-band. While it is well known that quantum dots based on several material systems can host isolated spin complexes with transition energies in the telecom bands, optical initialization and control of these has so far been outstanding. The presented results are not unexpected, but they are nevertheless an important step in the development of solid state spin systems for long-distance quantum network applications. As such I believe this manuscript will be suitable for publication in Nature Communications after my very few comments below have been addressed.

The manuscript is well written and the presented work supports the claims made by the authors. To my understanding the data analysis was performed carefully and leads nicely to the presented interpretation of the results. The methods and reference to earlier work are detailed enough for someone to reproduce these findings.

Comments:

- The authors give a very detailed overview of optically active solid state spin systems, and convincingly lay out the need for a spin photon interface with access to the telecom C-band. But very little reference is made to the efforts in their own groups as well as others developing quantum dots for the telecom bands, even though these works already clearly identify the presence of isolated optically addressable charge/spin carriers in such dots (eg Kettler et al Phys. Rev. B 94, 045303 (2016), Mrowinski et al PRB 100 115310 (2019), Ha et al Applied Physics Express 13 025002 (2020), Anderson et al Appl. Phys. Lett. 118, 014003 (2021)). It would be instructive to add a discussion of these efforts to the manuscript. In my view, the novelty in this work lies not in the proposition of an isolated spin system in a telecom quantum dot but in its control, and in addition to the discussion of previous works the text should also be amended to reflect this.

- Figure 3d: Here, the authors show that emission from the read-out transition is much more likely after the correct spin has been prepared. However, I am surprised that even after preparing the opposite spin, a non-negligible amount of light is scattered on the read-out transition. Looking at the intensities, emission from the read-out transition seems only about 6-8x more likely after the correct spin has been prepared. Is there an intuitive understanding of how this is compatible with a spin preparation fidelity of 96%? Also, what is the reason for the little extra peak about 7ns after the initialization step?

Minor comment:

- The title makes no sense grammatically. Maybe change to 'Optical Charge Injection and Full Coherent Control of a Spin-Qubit with optical transition wavelengths in the telecom C-band'?

Response Letter manuscript NCOMMS-21-25903-T

First, we would like to express our gratitude for the referees' very detailed reading of our work and their valuable comments. We followed all the suggestions and corrected the manuscript accordingly. The revised parts of the text are indicated in red in the main manuscript, and changes are also highlighted in the reply letter below for the referees' and editor's convenience. Please find below our detailed response to the referees' suggestions and comments.

Reviewer #1:

Solid-state spin qubits, which can be manipulated with C-band photons, have been expected as one of key elements for long-distance quantum networks. This report, demonstrating the all-optical coherent control of a spin-qubit in a quantum dot in the C-band, will provide a route towards the goal. The manuscript is well organized and describes the experiments the authors performed and the relevant discussions clearly.

However, based on the following facts, I think that the impact of the current work is not sufficiently strong as a paper in Nature Communications.

1. As the authors cite, coherent manipulation of spins in semiconductor QDs has been reported previously in many papers. Although the operation wavelengths are at around 1 μm , out of the C band, the fundamental schemes for the spin manipulation applied in the present work have been already discussed in the previous reports.

The coherent control of spins in InAs/GaAs semiconductor QDs emitting up to 1 μm has been indeed already reported by multiple groups. However, in the pursuit to realize a scalable quantum network, one of the most significant challenges is to extend the distance of quantum communication to a practically useful scale. As we stress out in the manuscript, access to spin-qubits with telecom band transitions provides a clear path towards this goal as an alternative to frequency-conversion approach. In this regard our work brings advances into mentioned fields in several aspects, as it introduces a new optically active solid-state spin-qubit system, interfaces it with telecom band photons, and finally demonstrates optical spin injection, initialization, read-out and complete quantum coherent control.

2. The technology enabling the demonstration, the crystal growth technology for C-band QDs on GaAs substrate, has been reported in ref [28].

Indeed, the growth of 1.55 μm compatible QDs on metamorphic layers has been previously reported. However, as the referees #2 and #3 stress out, the main objective and novelty of this manuscript lay down in utilization of such structures as spin-qubits and demonstration of the all-optical injection, initialization, manipulation, and read-out of isolated spin complexes with transition energies in the telecom bands, which was yet unaccomplished and highlight this system as particularly promising for long-distance quantum network applications.

On the other hand, I believe that it is worth reporting the sophisticated and high-quality experiments somewhere else. Other specific comments are listed below.

The more specific comments of reviewer #1 are addressed in a detailed fashion in the following.

1. In this study, hole spins are utilized for the demonstrations. Why did the authors choose X+ not X?

For our study we utilize a charged exciton (X^+) complex, which recombines radiatively into state with single carrier (hole in our case) trapped in QD. Such trapped single carrier possesses spin-degree of freedom, which we use as a qubit. In contrast, neutral exciton complex (X) does not have a non-zero spin ground state which could be utilized as long-lived qubit state.

Previous spectroscopic studies of the excitonic species in InAs/GaAs metamorphic QDs emitting in C-band showed that positively charged excitons are dominant transitions observable under non-resonant excitation conditions [36]. Since in the current generation of devices we do not have control over the charge type trapped in the QD, we naturally preselected positively charged excitons for our study. To clarify why X^+ complexes were used, we added the following sentence into the Methods section:

“In particular, positively charged trions formed by a single electron and two holes are identified as dominant transitions in agreement with previous theoretical and experimental studies [36]. Consequently, in this work, we focus on an isolated emitter with an optical transition between a positively charged exciton and a ground hole state (see Supplementary Information S2).”

[36] Carmesin, C. et al. Structural and optical properties of InAs/(In)GaAs/GaAs quantum dots with single-photon emission in the telecom C-band up to 77 K. *Phys. Rev. B* 98, 125407 (2018).

2. What are advantages to use hole spins instead of electron spins?

The main advantage of using hole spins instead of electron spins is the hole’s smaller hyperfine interaction strength with the nuclear spin environment due to the hole’s p-like symmetry and thus smaller carrier overlap at the nuclear sites. This is expected to allow achieving longer coherence times (currently limited by excitation induced dephasing) and will lay the ground for future quantum networks at telecom wavelength without the need for quantum frequency conversion. Also, given the predominantly heavy-hole ground state character, the hyperfine interaction is primarily concentrated along the growth axis and can be suppressed by a transverse external magnetic field.

We added the following sentence to page 2 of the manuscript:

“The benefit of using the hole state over the electron is hole’s smaller hyperfine interaction strength with nuclear spins due to the hole’s p-like symmetry [37,43] and thus potentially better hole coherence.”

[37] Brunner, D. et al. A Coherent Single-Hole Spin in a Semiconductor. *Science* (80-.). 325, 70–72 (2009).

[38] Fallahi, P., Yilmaz, S. T. & Imamoğlu, A. Measurement of a Heavy-Hole Hyperfine Interaction in InGaAs Quantum Dots Using Resonance Fluorescence. *Phys. Rev. Lett.* 105, 257402 (2010).

[39] Chekhovich, E. A., Krysa, A. B., Skolnick, M. S. & Tartakovskii, A. I. Direct Measurement of the Hole-Nuclear Spin Interaction in Single InP/GaInP Quantum Dots Using Photoluminescence Spectroscopy. *Phys. Rev. Lett.* 106, 027402 (2011).

[40] De Greve, K. et al. Ultrafast coherent control and suppressed nuclear feedback of a single quantum dot hole qubit. *Nat. Phys.* 7, 872–878 (2011).

[41] Greilich, A., Carter, S. G., Kim, D., Bracker, A. S. & Gammon, D. Optical control of one and two hole spins in interacting quantum dots. *Nat. Photonics* 5, 702–708 (2011).

[42] Godden, T. M. et al. Coherent Optical Control of the Spin of a Single Hole in an InAs/GaAs Quantum Dot. *Phys. Rev. Lett.* 108, 017402 (2012).

[43] Prechtel, J. H. et al. *Decoupling a hole spin qubit from the nuclear spins. Nat. Mater.* 15, 981–986 (2016).

Reviewer #2 (Remarks to the Author):

This paper presents a set of measurements on a new type of quantum-dot (QD) spin qubit, which naturally emits light in the telecom range. Historically, InAs QDs typically emit at shorter wavelengths, but emission into the telecom range is a major benefit for quantum networks. Some of the authors of the present work developed a new type of InAs QD that emits around 1550nm a few years ago, and the present paper shows that this system can be used as a spin qubit.

This paper demonstrates coherence of the charge and spin dynamics using a set of “conventional” measurements that are typical for regular InAs electron-spin QDs. On the whole, the data are convincing and, I think this work has the potential to be a major step forward.

We would like to thank the referee for his/her positive assessment and comments. We agree that this work is a major step towards quantum networks at telecom wavelengths with monolithic semiconductor technology without the need of frequency conversion.

I have two main comments for the authors:

1. At various points the authors mention the fidelities of different operations of their spin qubit, even though they have not performed state or process tomography. I think I understand their intended meaning, and I know that language of this sort may be common in the InAs QD literature, especially the older works. However, I think it best to reserve the term “fidelity” for cases when state or process tomography have been performed, especially in today’s world. I think the authors are not trying to claim record fidelities, so this should not be so challenging to fix. They could for example, just omit their claims of fidelities and leave the visibilities specified.

Following the referee suggestion, we amended the text accordingly and removed the fidelity terms.

2. The data presented show that the system works reasonably well as a qubit, but I was hoping they could convince me that it would be a really good qubit. Some questions that come to mind include:

- a. Do the authors think it is possible to achieve higher degrees of spin-photon entanglement (at telecom wavelengths) with their telecom QDs than with regular InAs QDs and wavelength converters?

This is an interesting question, which will be certainly answered in further studies on InAs/GaAs QDs based on a metamorphic buffer layer. At this point, we believe that in principle there is no reason why QDs emitting in the telecom regime should not reach spin-photon entanglement levels achievable for standard InAs QDs. In fact, we speculate that spin lifetime and coherence can potentially outperform regular InAs QDs, due to the presence of the strain relaxing layers. The lattice strain in QDs can influence the hole coherence by two effects: (i)

mixing the predominantly heavy-hole ground state with light-hole and (ii) enhancing the electric quadrupolar effects which play a dominant role in hole hyperfine interaction with nuclear spins. Both effects can be suppressed by decreasing the strain in the QD, which suggest that metamorphic QD could be a potentially better qubit than regular InAs QD. Some further work is required to establish the validity of those assumptions.

The following text has been added to the summary section:

“While the intrinsic coherence limitations of the spin system investigated in this work remain to be revealed in further studies, the presence of strain-relaxed layers suggest that it could outperform the typical well-established InAs/GaAs QDs. It was shown that reducing the lattice strain in semiconductor QDs increases the spin coherence by limiting the electric quadrupolar effects which play a dominant role in hole hyperfine interaction with nuclear spins [43,44]. In the case of hole qubits, less strain translates also into smaller light-hole admixture, which was shown to limit the spin-hole lifetime [45,46] and coherence time [44].”

[43] Bulutay, C. *Quadrupolar spectra of nuclear spins in strained InGaAs quantum dots.* *Phys. Rev. B* 85, 115313 (2012).

[44] Urbaszek, B. *et al.* *Nuclear spin physics in quantum dots: An optical investigation.* *Rev. Mod. Phys.* 85, 79–133 (2013).

[45] Krzykowski, M., Gawarecki, K. & Machnikowski, P. *Hole spin-flip transitions in a self-assembled quantum dot.* *Phys. Rev. B* 102, 205301 (2020).

[46] Zhai, L. *et al.* *Low-noise GaAs quantum dots for quantum photonics.* *Nat. Commun.* 11, 4745 (2020).

b. What challenges do they expect to encounter, and how could these be overcome?

In the current study, we perform a charge injection using non-resonant optical ps pulses. We believe that next-generation samples would require a more deterministic charge injection scheme. In regular InAs QDs, embedding the QDs in an n-i-p diode allows (i) trapping the charge state by Coulomb blockade; (ii) reducing the charge noise; and (iii) tuning the transition frequency via a gate voltage. While QDs based on a metamorphic buffer layer are currently at a slightly different stage of development than regular InAs QDs, we believe that this technology could be adapted to our structures. Furthermore, we plan on integrating our QDs with optical cavities to improve the photon collection efficiency and the spin-coherence/photon lifetime ratio.

c. Does the hole nature of this qubit confer any advantages (or disadvantages) with respect to electron-spin QDs?

The main advantage of using hole spins instead of electron spins is the hole’s smaller hyperfine interaction strength with the nuclear spin environment due to the hole’s p-like symmetry and thus smaller carrier overlap at the nuclear sites. This is expected to allow achieving longer coherence times (currently limited by excitation induced dephasing) and will lay the ground for future quantum networks at telecom wavelength without the need for quantum frequency conversion. Also, given the predominantly heavy-hole ground state character, the hyperfine interaction is primarily concentrated along the growth axis and can be suppressed by a transverse external magnetic field.

We added the following sentence to page 2 of the manuscript:

“The benefit of using the hole state over the electron is hole’s smaller hyperfine interaction strength with nuclear spins due to the hole’s p-like symmetry [37,43] and thus potentially better hole coherence.”

- [37] Brunner, D. et al. A Coherent Single-Hole Spin in a Semiconductor. *Science* (80-.). 325, 70–72 (2009).
- [38] Fallahi, P., Yilmaz, S. T. & Imamoğlu, A. Measurement of a Heavy-Hole Hyperfine Interaction in InGaAs Quantum Dots Using Resonance Fluorescence. *Phys. Rev. Lett.* 105, 257402 (2010).
- [39] Chekhovich, E. A., Krysa, A. B., Skolnick, M. S. & Tartakovskii, A. I. Direct Measurement of the Hole-Nuclear Spin Interaction in Single InP/GaInP Quantum Dots Using Photoluminescence Spectroscopy. *Phys. Rev. Lett.* 106, 027402 (2011).
- [40] De Greve, K. et al. Ultrafast coherent control and suppressed nuclear feedback of a single quantum dot hole qubit. *Nat. Phys.* 7, 872–878 (2011).
- [41] Greilich, A., Carter, S. G., Kim, D., Bracker, A. S. & Gammon, D. Optical control of one and two hole spins in interacting quantum dots. *Nat. Photonics* 5, 702–708 (2011).
- [42] Godden, T. M. et al. Coherent Optical Control of the Spin of a Single Hole in an InAs/GaAs Quantum Dot. *Phys. Rev. Lett.* 108, 017402 (2012).
- [43] Prechtel, J. H. et al. Decoupling a hole spin qubit from the nuclear spins. *Nat. Mater.* 15, 981–986 (2016).

d. What about the yield of their QDs with respect to regular InAs QDs?

Based on the previous spectroscopic studies of InAs metamorphic QDs we expect photoluminescence yield similar to regular InAs QDs [see [36] Carmesin, C. et al. Structural and optical properties of InAs/(In)GaAs/GaAs quantum dots with single-photon emission in the telecom C-band up to 77 K. *Phys. Rev. B* 98, 125407 (2018)].

e. It seems like the frequency shifts between the H and V lines (Fig. 1 d) are smaller than in conventional InAs QDs—does this make a difference to the qubit performance, especially in cases where readout fidelity is important, like teleportation?

Splitting between all the transitions in the double lambda QD system is directly proportional to the g-factors of the ground and the excited state of the trion. Crucial for the qubit operation is ground state splitting (related to transitions 1-3 or 2-4), which in the case of our QDs is relatively high, as the ground state in-plane g-factor yields a value of $|2.7|$ in comparison to $|0.1-0.6|$ usually observed for electrons and holes in regular InAs QDs [see for instance Bennett, A., Pooley, M., Cao, Y. et al. Voltage tunability of single-spin states in a quantum dot. *Nat Commun* 4, 1522 (2013)]. Such splitting allows using smaller magnetic fields than in regular InAs QDs and allows for easier spectral addressing of H and V transitions within the same lambda system. On the other hand, transitions 1-2 and 3-4 related to two V-systems are indeed closely spaced spectrally at small fields, which requires using polarization for addressing specific transitions. While access to the V-system is not required for the coherent control or generation of spin-photon entanglement, it might be advantageous in some cases for read-out protocols. In terms of qubit performance, we believe that small splitting between 1-2 and 3-4 should not play a role.

Reviewer #3 (Remarks to the Author):

Dusanowski et al describe the first demonstration of spin control and manipulation with an optical interface in the telecom C-band. While it is well known that quantum dots based on several material systems can host isolated spin complexes with transition energies in the telecom bands, optical initialization and control of these has so far been outstanding. The

presented results are not unexpected, but they are nevertheless an important step in the development of solid state spin systems for long-distance quantum network applications. As such I believe this manuscript will be suitable for publication in Nature Communications after my very few comments below have been addressed.

The manuscript is well written and the presented work supports the claims made by the authors. To my understanding the data analysis was performed carefully and leads nicely to the presented interpretation of the results. The methods and reference to earlier work are detailed enough for someone to reproduce these findings.

We would like to thank the referee for his/her positive assessment.

Comments:

- The authors give a very detailed overview of optically active solid state spin systems, and convincingly lay out the need for a spin photon interface with access to the telecom C-band. But very little reference is made to the efforts in their own groups as well as others developing quantum dots for the telecom bands, even though these works already clearly identify the presence of isolated optically addressable charge/spin carriers in such dots (eg Kettler et al Phys. Rev. B 94, 045303 (2016), Mrowinski et al PRB 100 115310 (2019), Ha et al Applied Physics Express 13 025002 (2020), Anderson et al Appl. Phys. Lett. 118, 014003 (2021)). It would be instructive to add a discussion of these efforts to the manuscript. In my view, the novelty in this work lies not in the proposition of an isolated spin system in a telecom quantum dot but in its control, and in addition to the discussion of previous works the text should also be amended to reflect this.

Following the referee suggestion we added an additional paragraph in the introduction, describing efforts towards identifying optically active spin carriers in telecom emitting QDs:

“Alternative systems that could potentially allow interfacing spins with telecom-wavelength photons are InAs QDs relying on InP substrate [28-30] or strain-relaxed layers in GaAs [31-33]. Spectroscopic studies of such systems identified already charged exciton complexes with intrinsic optical transitions in the C-band offering access to isolated electron and hole states [29,30,33,34].”

[28] Mrowiński, P. et al. Magnetic field control of the neutral and charged exciton fine structure in single quantum dashes emitting at 1.55 μm . *Appl. Phys. Lett.* 106, 053114 (2015).

[29] Ha, N. et al. Single photon emission from droplet epitaxial quantum dots in the standard telecom window around a wavelength of 1.55 μm . *Appl. Phys. Express* 13, 025002 (2020).

[30] Anderson, M. et al. Coherence in single photon emission from droplet epitaxy and Stranski–Krastanov quantum dots in the telecom C-band. *Appl. Phys. Lett.* 118, 014003 (2021).

[31] Kettler, J. et al. Neutral and charged biexciton-exciton cascade in near-telecom-wavelength quantum dots. *Phys. Rev. B* 94, 045303 (2016).

[32] Mrowiński, P. et al. Excitonic complexes in MOCVD-grown InGaAs/GaAs quantum dots emitting at telecom wavelengths. *Phys. Rev. B* 100, 115310 (2019).

[33] Nawrath, C. et al. Coherence and indistinguishability of highly pure single photons from non-resonantly and resonantly excited telecom C-band quantum dots. *Appl. Phys. Lett.* 115, 023103 (2019).

[34] Rudno-Rudziński, W., Burakowski, M., Reithmaier, J. P., Musiał, A. & Benyoucef, M. Magneto-Optical Characterization of Trions in Symmetric InP-Based Quantum Dots for Quantum Communication Applications. *Materials* 14, 942 (2021).

- Figure 3d: Here, the authors show that emission from the read-out transition is much more likely after the correct spin has been prepared. However, I am surprised that even after preparing the opposite spin, a non-negligible amount of light is scattered on the read-out transition. Looking at the intensities, emission from the read-out transition seems only about 6-8x more likely after the correct spin has been prepared. Is there an intuitive understanding of how this is compatible with a spin preparation fidelity of 96%? Also, what is the reason for the little extra peak about 7ns after the initialization step?

Without optical pumping the probability of the spin being prepared in the up or down state is 50%. Such configuration corresponds to initial emission intensity I_0 . By preparing the spin with 75% ($=0.5+0.5/2$) probability in the spin-up state (one spin-pumping cycle), emission from transition 1 will be increased to $1.5I_0$ and emission from transition 2 reduced to $0.5I_0$. The probability of preparing the given spin-up state P_{up} by optical pumping will induce $2(1 - P_{up})I_0$ intensity on the spin-down state and $2P_{up}I_0$ intensity on the spin-up state. Inversely, from the change of intensity C one can estimate the probability of preparing the spin in the up state: $P_{up} = 1 - C/2$, and spin-down state $P_{down} = C/2$. For example for the change of intensity by a factor of x8, the probability of preparing spin in a given state is equal to $1-1/2*(1/8)= 15/16=93.75\%$. The exact dynamics and probability of the spin initialization is captured in the rate equation model presented in Supplementary Information Section 4 (page 5).

The little extra peak at 7 ns is an experimental artefact related to non-ideal OPO laser system alignment, resulting in presence of very low-intensity extra pulses at ± 3 ns delay from the original pulse sequence at 1550 nm. Despite the low intensity of those pulses, they do slightly excite the QD, which is visible in a form of decay as noticed by the reviewer. To clarify this point we added the following sentence into the Methods section:

“In Fig.3d there are very small intensity peaks visible at 2.5 ns and 7.7 ns time. Those artefacts are related non-ideal 76 MHz readout laser pulse sequence, consisting of the main pulse and two extra satellite pulses at ± 3 ns with around two orders of magnitude lower intensity.”

Minor comment:

- The title makes no sense grammatically. Maybe change to 'Optical Charge Injection and Full Coherent Control of a Spin-Qubit with optical transition wavelengths in the telecom C-band'?

Following the referee suggestion, we changed the manuscript title. Taken the 15 words title limit we propose the following:

“Optical Charge Injection and Coherent Control of a Quantum Dot based Spin-Qubit at Telecom Wavelength”

REVIEWERS' COMMENTS

Reviewer #1 (Remarks to the Author):

I have carefully read the authors' response letter and the revised manuscript.

The authors addressed all the reviewers' comments satisfactorily, convincing me of the importance and the significance of the present work in the field.

Now I recommend the manuscript for publication as is.

Reviewer #2 (Remarks to the Author):

The authors have done a nice job responding to the comments from the reviewers. I recommend publication. The following points may optionally be addressed.

I had intended my question d. to refer to fabrication yield instead of photoluminescence yield, but I suppose the latter quantity is also important.

If the authors felt like including some of the text from their answer to b. in the main text, that might be instructive for the reader (unless they feel this would be giving away too much).

The new proposed title may still be not quite right. An alternative could be "Optical Charge Injection and Coherent Control of a Quantum-Dot Spin-Qubit emitting at Telecom Wavelengths"

Reviewer #3 (Remarks to the Author):

I thank the authors for addressing my comments and I am satisfied with the changes they made to the manuscript. As such I can now recommend this work for publication in Nature Communications.

RESPONSE TO REVIEWERS' COMMENTS - NCOMMS-21-25903-T

Please find below our detailed response to the referees' suggestions and comments. The revised parts of the text are indicated in red in the main manuscript, and changes are also highlighted in the reply letter below for the referees' and editor's convenience.

Reviewer #1 (Remarks to the Author):

I have carefully read the authors' response letter and the revised manuscript. The authors addressed all the reviewers' comments satisfactorily, convincing me of the importance and the significance of the present work in the field. Now I recommend the manuscript for publication as is.

We would like to thank the referee for his/her positive assessment.

Reviewer #2 (Remarks to the Author):

The authors have done a nice job responding to the comments from the reviewers. I recommend publication. The following points may optionally be addressed.

We would like to thank the referee for his/her positive assessment.

I had intended my question d. to refer to fabrication yield instead of photoluminescence yield, but I suppose the latter quantity is also important.

Within current work we are using quantum dot emitters formed in self-assembled manner using metalorganic vapour phase epitaxy (MOVPE) on GaAs substrates. In this approach quantum dots are formed through the strain build up due to the lattice constants difference between InAs and InGaAs metamorphic buffer (so called Stranski–Krastanov growth mode). Areal density of the QDs can be estimated using AFM just after QD layer is grown (in case of our sample estimated to be around $6 \times 10^9 \text{ cm}^{-2}$). However, for full structure, where QDs are overgrown with InGaAs cap, final QD density cannot be estimated by this technique anymore. Capping layer growth step (necessary to inhibit surface recombination) can influence the number and size of the QDs. For comparison micro-PL experiments estimate the area density of optically active QDs on the level of 10^7 cm^{-2} . In this context we could define the fabrication yield of single optically active QD for a given wafer growth run. Assuming that number of QD does not change after capping, the effective number of optically active QDs will correspond to around 0.16%. The exact details regarding sample fabrication, QD size distribution and density can be found in Appl. Phys. Lett. 111, 033102 (2017). (already referenced in Methods section)

If the authors felt like including some of the text from their answer to b. in the main text, that might be instructive for the reader (unless they feel this would be giving away too much).

Following the referee suggestion, the following piece of text have been added into summary paragraph:

“Thanks to mature GaAs fabrication technology, our QD-spin system could be further combined with cavities and other photonic structures. This in turn might allow increasing the photon collection efficiency, speed up the spin-initialization process and achieve high optical transition cyclicity [51]

for single-shot read-out. Further performance improvement could be achieved by utilizing an n-i-p diode structure, which was shown to allow trapping the charge state via Coulomb blockade, tuning the emission frequency using gate voltage and reducing the charge noise [50].”

[50] L. Zhai, M. C. Löbl, G. N. Nguyen, J. Ritzmann, A. Javadi, C. Spinnler, A. D. Wieck, A. Ludwig, and R. J. Warburton, Low-Noise GaAs Quantum Dots for Quantum Photonics, Nat. Commun. 11, 4745 (2020).

[51] M. H. Appel, A. Tiranov, A. Javadi, M. C. Löbl, Y. Wang, S. Scholz, A. D. Wieck, A. Ludwig, R. J. Warburton, and P. Lodahl, Coherent Spin-Photon Interface with Waveguide Induced Cycling Transitions, Phys. Rev. Lett. 126, 013602 (2021).

The new proposed title may still be not quite right. An alternative could be “Optical Charge Injection and Coherent Control of a Quantum-Dot Spin-Qubit emitting at Telecom Wavelengths”

Following the referee suggestion, we changed the manuscript title to: “Optical Charge Injection and Coherent Control of a Quantum-Dot Spin-Qubit emitting at Telecom Wavelengths”.

Reviewer #3 (Remarks to the Author):

I thank the authors for addressing my comments and I am satisfied with the changes they made to the manuscript. As such I can now recommend this work for publication in Nature Communications.

We would like to thank the referee for his/her positive assessment.